# Incidence and Risk Factors for Acute Articular Cartilage Tears in Military and Other Occupational Settings: A Systematic Review

**DOI:** 10.3390/healthcare12050595

**Published:** 2024-03-06

**Authors:** Kristy Robson, Rodney Pope, Robin Orr

**Affiliations:** 1Three Rivers Department of Rural Health, Charles Sturt University, Albury, NSW 2640, Australia; 2School of Allied Health, Exercise and Sports Sciences, Charles Sturt University, Albury, NSW 2640, Australia; rpope@csu.edu.au; 3Tactical Research Unit, Bond University, Robina, QLD 4226, Australia; rorr@bond.edu.au; 4Faculty of Health Science and Medicine, Bond University, Robina, QLD 4226, Australia

**Keywords:** occupational risk factors, incident rates, work injury, joint injury

## Abstract

Damage to the articular cartilage resulting in an acute tear can lead to functional changes within the joint and increase the risk of osteoarthritis developing. There is limited understanding of the association between occupational risk factors and sustaining an acute articular cartilage tear in the military and other physically demanding occupations. Therefore, the aim of this systematic review was to identify and evaluate original research reporting on occupational risk factors associated with sustaining acute articular cartilage tears. Methods: A systematic review following the Preferred Reporting Items for Systematic review and Meta-Analysis—Protocols was conducted and registered with the Open Science Framework. Key academic databases were searched using terms from the following concepts: risk or cause, paid occupations, and acute articular cartilage tears. Results: Of an initial 941 studies, 2 studies met the eligibility criteria, both reporting data from military contexts; only one evaluated acute articular cartilage tears in both males and females. One paper focused on articular cartilage injury within the knee and the other within the ankle joint with incidence rates being 0.2 and 0.3 per 1000 person-years, respectively. People in more physically active occupations and individuals with an above-normal body mass index were reported as being at higher risk of sustaining an acute articular cartilage tear. Conclusion: Physically demanding occupations, such as the military, may increase the risk for acute tears of the articular cartilage. However, the findings of this review indicate there is a paucity of research to underpin understanding of the injury mechanisms and occupational risk factors for acute articular cartilage tears.

## 1. Introduction

Articular cartilage within a joint provides a low-friction gliding surface while also acting as a shock absorber and reducing peak pressures to the subchondral bone [1]. An acute articular cartilage tear is an injury involving tearing or damage of the articular (hyaline) cartilage of a joint, resulting in a sudden onset of pain or tenderness [2]. These injuries can occur due to trauma from significant mechanical stresses or forces to the joint [3], which are often the result of compression, shear, or tension forces [2] and can be associated with occupational activities within physically demanding professions such as the military, construction and building industries, and mining [4,5,6].

Disruption to the articular cartilage such as those seen in acute articular tears can further lead to functional changes within the joint as well as increasing the risk of osteoarthritis [7,8]. It is not uncommon to see acute articular cartilage damage alongside other injuries such as ligamentous tears or synovial joint damage, including meniscal and joint capsule injuries [9,10]. An acute articular cartilage tear can be associated with acute knee haemarthrosis [11] or identified during intra-articular surgery on an affected joint [12,13]. Some studies have identified that a delay in anterior cruciate ligament reconstruction surgery can also lead to acute articular damage due to instability within the knee joint resulting in greater shear forces [14,15].

Damage to the articular cartilage is typically seen in weight-bearing joints [16], with articular cartilage lesions caused by a traumatic, noncontact mechanism of injury—as reported in 32–65% of patients presenting for arthroscopy [11,17,18]. These types of injuries can also be a result of high-impact sporting activities [11,19]. Sports-related movements, commonly seen in soccer, basketball, football, and high-intensity strength and conditioning programs, have been associated with traumatic injury to the articular cartilage [20,21,22,23]. Occupations that involve twisting, kneeling, repetitive stress, or indirect loading of joints can also increase the risk of acute articular tears [24,25]. Other studies have also highlighted that other joints that are subject to high levels of stress, such as the elbow, may be at risk for cartilage injury [26].

While injury to the articular cartilage is often seen in younger, active populations and associated with sporting activities [27,28], limited studies have focused on occupational risks associated with isolated damage to articular cartilage. Brady et al. [29] identified an increased prevalence of trochlear lesions in firefighters undergoing knee arthroscopy for an acute meniscal or anterior cruciate ligament tear, which may be due to the physical demands of this type of occupation. However, it is difficult to determine whether the chondral lesions were a result of a traumatic injury alongside the acute concomitant injuries or a result of chronic micro-trauma. Shaw et al. [30] used fresh juvenile porcine stifle joints to assess the effects of blast exposure, commonly seen in military operational settings, on articular cartilage, finding that chondrocyte death occurred after exposure to a simulated blast wave. Other studies have focused on the incidence of acute articular cartilage injury in the knee and ankle within a military setting [31,32].

Due to the high rates of concomitant injuries associated with acute articular cartilage tears reported in the literature [33,34,35], there is limited understanding of the specific risk factors associated with isolated events of acute articular cartilage injury, particularly in occupational settings, including non-sporting and non-military occupations as well as in highly physical occupations such as the military. Therefore, the aim of this systematic review was to identify the aetiology, incidence, and known risk factors for workers sustaining acute articular cartilage tears in paid occupational settings. For the purpose of this review, acute articular cartilage tears exclude chronic or degenerative tears of articular cartilage.

## 2. Methodology

A systematic review, using a systematic approach and critical narrative approach to synthesis, was conducted to identify and synthesise findings from published studies that investigated factors that are associated with risk of acute articular cartilage tears in occupational settings. The design and reporting of the review was guided by the Preferred Reporting Items for Systematic review and Meta-Analysis—Protocols (PRISMA-P) [36].

### 2.1. Protocol and Registration

The project and the protocol for this systematic review were registered with the Open Science Framework on 28 July 2020 (https://osf.io/3u9eg) [37].

### 2.2. Eligibility Criteria, Data Sources, and Search Terms

A systematic search of key databases was completed in March 2023. The databases concerned, PubMed, Elton B Stevens Company (EBSCO, including SPORTDiscus and Cumulative Index to Nursing and Health Care Literature [CINAHL]), and ProQuest, were searched using search terms derived from three themes: ‘acute articular cartilage tear’, ‘work’, and ‘risk’. An example search string for the PubMed database can be found in Table 1.

### 2.3. Inclusion and Exclusion Criteria

The inclusion criteria for this systematic review were: (a) quantitative study design, (b) study was conducted in humans 16 years or older or provided extractable data for these age groups, (c) published in the English, Portuguese, French, Italian, or Spanish languages (which the reviewer team could read and translate into English), (d) investigated factors, or exposures, or hazards, or causes, or mediators associated with the development or prevention of acute articular cartilage tears in personnel engaged in paid occupations, or the incidence, or prevalence, or likelihood of the condition occurring in particular occupational groups, and (e) used diagnostic criteria consistent with (or similar to) the Australian Government Repatriation Medical Authority [38] (RMA) Statement of Principles (SoP) meaning of ‘acute articular cartilage tear’, which was: (i) injury involving tearing or damage of the articular (hyaline) cartilage of a joint resulting in a sudden onset of pain and tenderness; (ii) excluding chronic tears and degenerative tears of articular cartilage; and (iii) presentation may be accompanied by swelling, bruising, or loss of functional ability, occurring within the 24 h following the injury.

Articles were excluded if they were: (a) a literature review of any type, (b) only a published abstract, (c) a non-peer-reviewed article, (d) not original research, (e) a qualitative study, (f) a study of pharmacological interventions or ergogenic aids, (g) a study of unpaid athletes or people in volunteer occupations, (h) a study that investigated populations with known pre-existing medical conditions, (i) a prospective study focusing on older populations, or (j) a research protocol.

### 2.4. Screening and Selection

All titles and abstracts of articles identified through the systematic search strategy were screened by one reviewer (KR), and duplicates and articles that clearly did not meet the inclusion criteria were removed. Full-text copies of all of the remaining studies were obtained and reviewed independently by two reviewers (KR and RP) to determine eligibility for inclusion, based on the inclusion and exclusion criteria. Reasons for exclusion of full-text articles were documented and any disagreements were resolved through discussion and subsequent consensus. The results of the search, screening, and selection processes were documented in a PRISMA flow chart [36].

### 2.5. Level of Evidence and Methodological Quality Assessment

Each study was evaluated against the levels of evidence described by the National Health and Medical Research Council (NHMRC) Evidence Hierarchy scale [39]. Using this approach studies are ranked on a scale from Level I to IV, with Level I indicating the research design with the highest level of evidence to answer a particular type of research question. While Levels I, II, and IV are single points on the scale, Level III has three subscales (III-1 [higher] to III-3 [lower]) [39]. For this systematic review, the NHMRC scale was selected rather than the Grading of Recommendations Assessment, Development, and Evaluation (GRADE) system of rating quality of evidence and grading strength of recommendations in systematic reviews. This was because the GRADE system is not designed for assessing evidence associated with studies that focus on answering research questions about risk or prognosis [40]. However, noting the limitations of the NHMRC Evidence Hierarchy scale, particularly the fact it does not consider the methodological quality of individual studies with specific designs [39,41], each study included in this systematic review was also assessed for risk of bias and critically appraised for methodological quality, using the Critical Appraisal Skills Programme (CASP) toolkit and particularly the tool for assessing cohort studies [42]. These tools contain a range of questions aimed at guiding the reviewer through assessing the validity, relevance, objectives, methodology, results, and discussion of each study.

The CASP cohort study checklist [42] incorporates 12 questions. The first two questions consider whether the study had a clear focus and whether sampling and recruitment of the cohort were conducted appropriately. The following seven questions consider accuracy of measurements, risk of confounding, follow-up of participants, precision of results, and overall risk of bias and validity of findings. The final three questions consider the external validity and implications of the findings in the local context and so were not considered in our critical appraisals, due to their context-specificity.

For each question considered in the checklist, one point was awarded if the reviewer answered ‘yes’, and zero points were awarded if the reviewer answered ‘no’ or ‘can’t tell’. Some questions within the checklist could not be answered numerically but informed scores assigned in Question 9. As noted above, Questions 10–12 were removed from the scoring as these related to the applicability of results to the local context.

The questions were then rated on a binary scale and study quality scores were converted to percentage scores [43] and then classified as ‘poor’ (<45.4%), ‘fair’ (45.4–61.0%), or good (>61.0%) in an approach used in previous research [44]. Additional narrative commentary was also provided on the specific study limitations. In this way, a combination of the levels of evidence and methodological quality assessments was used to classify, appraise, and grade the evidence presented within each of the studies.

### 2.6. Data Extraction and Synthesis

Data relating to study characteristics and key findings were extracted from each included article and tabulated. Study characteristics extracted from each article were authors and year of publication, research design, research setting, participant details including population and sex and age profiles, study aim, and key outcome measures of relevance to the aims of the systematic review. All key findings from the included studies that were of relevance to addressing the aim of the review were also extracted.

Following data extraction, key findings from the included studies were synthesised using a critical narrative approach, in which the methodological quality and limitations of each included study were considered. Meta-analysis was not undertaken due to the relatively low number and heterogeneity of included studies.

## 3. Results

The PRISMA diagram outlining the results of the search, screening, and selection processes can be seen in Figure 1. The systematic database search yielded a total of 941 articles. Once duplicates were removed and the screening of titles and abstracts completed, 51 articles remained to be reviewed in full text. Assessment of these full-text articles against the study eligibility criteria resulted in two articles being included in the review, with 49 papers being excluded on the following basis: (a) participants having an existing medical condition, (b) the paper being a review, (c) the diagnosis of articular cartilage tear being inconsistent with the meaning of an acute articular cartilage tear specified in RMA SoPs [38] because it was not evident included injuries were acute, (d) the study was not conducted on live humans, (e) the paper was a conference abstract, (f) the study not stating whether the participants’ injuries were a result of paid employment, or (g) the study was a quantitative study (see Table 1).

Many studies identified in the initial search were excluded during the selection process because they were focused on chronic tears and degenerative tears of articular cartilage rather than on acute articular cartilage tears, which were the focus of this review. A number of those excluded studies involved participants with a pre-existing anterior cruciate ligament (ACL) or meniscal injury and investigated degenerative changes occurring in the articular cartilage subsequent to ACL or meniscal injuries. In addition, four studies [30,45,46,47] were excluded because they were undertaken on either cadavers or non-human simulated joints. One study of cadavers evaluated different surgical techniques for ACL reconstruction to identify the risk of chondral injury associated with each technique [45], and another measured the response and injury tolerance of articular cartilage during significant loading of the ankle joint similar to loading that would be applied in a vehicle crash [46]. One of the non-human simulated joint studies evaluated the impacts of shear and axial loads [47], and the other, the degree of injury to articular chondrocytes after exposure to simulated blast overpressure waves [30].

The two included studies were cohort studies [31,32], providing evidence at NHMRC level III-2. Despite some methodological limitations, both studies scored eight out of a possible nine for methodological quality. The limitations noted for both studies [31,32] included they did not clearly indicate how the diagnosis was determined, and possible confounding factors were not considered, such as previous injury for both studies and body mass index for one study [32]. Taken together, the critical appraisal results indicate the included studies were of high methodological quality for retrospective cohort designs.

The characteristics of the two included studies are presented in Table 2 [31,32]. Both were undertaken in military contexts, one in the United States [31] and the other in Finland [32]. Only one study [31] evaluated acute articular cartilage tears in both males and females. The other [32] involved only male participants. Key findings from the included studies that were of relevance to this review are listed in Table 3 and are further considered in Figure 2 and the synthesis that follows.

Each of the studies included in this review had a focus on military settings. The incidence rate for acute articular cartilage tears was assessed for a different joint in each of these cohort studies, which identified similar rates for the ankle and knee. The unadjusted incidence rate was 0.27 acute articular cartilage tears per 1000 person-years in the ankle [31] and 0.2 acute articular cartilage tears per 1000 person-years in the knee [32].

Analysis of acute articular cartilage injuries by age and sex was only undertaken in one of the studies [31], which found that females had a higher incidence rate than males for osteochondral lesions of the talus (OCLT), with an adjusted incidence rate ratio of 1.34 (95% CI 1.23–1.47; adjusted for race, age, rank, and branch of military service) when females were compared to males. The unadjusted incidence rate for males in this study was 0.27 acute articular cartilage tears (OCLT; affecting the ankle joint) per 1000 person-years, which was a similar finding to that in the study undertaken by Kuikka et al. [32], who identified an incidence rate of 0.2 hospitalised acute articular cartilage tears affecting the knee joint per 1000 person-years in their study of an all-male population.

Only one of the studies [31] reported incidence rates for acute articular cartilage tears by age category. That study found a higher adjusted incidence rate in older military personnel, with an adjusted injury risk ratio of 2.89 (95% CI 2.40–3.46) for the 35–39 years age category and 3.34 (95% CI 2.76–4.05) for the 40+ years age category when each was compared to younger personnel aged less than 20 years (Table 3 and Figure 2).

There were a number of findings in the included studies that were unique to a military setting. One study [31] found that an increased risk of acute articular cartilage injury (OCLT) was associated with service groups such as the Army and Marines compared to the Air Force and Navy after adjusting for confounders (Table 3). This study also found that white race and other race categories of military personnel had a higher risk of acute articular cartilage tears after adjusting for confounders compared to personnel categorised as being of black race (Table 3). Additionally, this study identified that junior and senior enlisted personnel were at higher risk of experiencing an OCLT compared to both junior officers and senior officers after adjusting for confounders (Table 3).

Only one of the included studies examined the association between training time and risk of acute articular cartilage injury. Kuikka et al. [32] found that 70% of hospitalised acute articular cartilage tears affecting the knee occurred within the first six months of a 12-month training period.

None of the included studies in this review investigated specific occupational tasks undertaken at the times when acute articular cartilage tears occurred. This limits the opportunity to determine particular occupational risk factors associated with this type of injury.

## 4. Discussion

The aim of this systematic review was to identify the aetiology, incidence, and known risk factors for workers sustaining acute articular cartilage tears in paid occupational settings. Despite the paucity of studies in this area that focused on occupational settings, key findings from this review indicate that more heavily physically active military populations such as Army and Marine personnel may be at a higher risk of sustaining an acute articular cartilage tear than those in Air Force and Navy occupations. In addition, factors including older age, service as enlisted personnel (rather than as officers), and being within the first six months of initial training were all associated with increased risk of acute articular cartilage injury in military personnel [31,32].

Accurately determining the incidence of acute articular cartilage injury in the general population and other occupations is difficult due to the absence of epidemiological studies in this area. Most studies that provide any incidence or prevalence data for articular cartilage injuries fail to distinguish between acute and chronic articular cartilage injuries [28,29,48,49,50]. No studies could be found that determined the overall incidence rate for acute articular cartilage tears within the general population, and, therefore, it is not possible to compare the incidence rates observed in the military populations considered in this review to those of the general population. However, Hjelle et al. [34] found in a study of a series of 1000 knee arthroscopies in Sweden, traumatic focal chondral lesions were identified in 28% of the participants. Similarly, Aroen et al. [33] found in a study of 1005 knee arthroscopies performed over a 6-month period in Norway that 20% identified a localised articular cartilage lesion without any degeneration noted. Bikash et al. [51], in their study of 75 patients undergoing knee arthroscopy in Nepal, found chondral lesions in 49% of cases. The latter study by Bikash et al. [51] had much smaller numbers compared to the previous two studies [33,34] and over 56% of patients were aged 21–30 years, which may explain the higher prevalence identified, as younger populations have been shown to have a higher prevalence [35].

Despite the challenges associated with quantifying how common acute articular cartilage tears are within the general population and specific occupations, findings from this review indicated that more active military occupations are at a higher risk of acute articular cartilage injury [31]. This finding is supported by findings of Widuchowski et al. [11], in a Polish general population study, indicating increased physical activity was associated with an increased risk of acute articular cartilage tear, and that 45% of acute articular cartilage tears were associated with undertaking sport, with the majority being a direct result of injury. The prevalence of focal chondral defects has also been shown to be higher in athletes (36%) than in the general population (16%) [52]. Additionally, a study undertaken by Majewski et al. [53], investigating injuries presenting to a German sports medicine clinic over a 10-year period, found that 11% of injuries involved articular cartilage damage, in individuals undertaking physical activity such as sport.

Age was highlighted as a potential risk factor for military personnel within this review, with Orr et al. [31] finding older active military personnel to be at a higher risk of acute articular cartilage injury than their younger counterparts. This finding is supported by the study of Yuksel et al. [28], who found that in a Turkish military setting, the risk of chondral lesions accompanying ACL tears was twice as high in patients aged 30 years and above compared to those under the age of 30 years. However, within the broader population, available evidence does not support this finding. Aroen et al. [33] found that localised cartilage lesions identified through arthroscopy were found more commonly in the younger age groups, with those in the 20–25 age range demonstrating the highest prevalence. This finding is also supported by a large study of the general population undertaken by Widuchowski et al. [11], who found that the largest age group of patients presenting with a chondral lesion, identified through arthroscopy in Poland, was the 21–30 years category. It is possible that the continuing physical demands imposed on older military personnel, which may not be a requirement for many older people in the general population, may explain these contrasting findings in these different populations.

Within this review, only one study investigated the gender of military personnel as a potential risk factor for acute articular cartilage tears and it found females to be at higher risk of acute articular cartilage injury [31]. In contrast, a study undertaken by Widuchowski et al. [11] identified that chondral lesions were diagnosed in 66% of males compared to 34% of females in their study of a general population. It should be noted that this study [11] combined all chondral lesions in its analyses, including both acute and chronic injuries of the cartilage; however, over 70% were reported to have had a traumatic onset. Additionally, a large study by Loes et al. [20] investigating knee injuries in Swiss youth sports found there was no difference in the prevalence of chondral injuries between genders. It is, therefore, possible that the experiences of military women and men in terms of relative rates of acute articular cartilage injuries are different from the experiences of women and men in other populations. Reasons for this require further investigation but may be associated with average differences between the sexes in levels and types of physical activity.

With respect to military rank, Orr et al. [31] found that enlisted personnel had a greater risk of sustaining an acute articular cartilage tear than officers. This finding is supported by the study of Lauder et al. [54], who found that enlisted military personnel had the highest overall hospitalisation incidence rate for all sports and physical training injuries combined. Why this is the case is unclear, but it could possibly be due to the requirement for lower-ranked personnel to undertake more frequent arduous physical activities compared to higher-ranked personnel [54]. This supposition is supported by Roy et al. [55], who suggest that for each step up in rank, injury decreases by 14%.

In this review, no studies could be found that identified a specific activity related to the development of an acute articular cartilage tear. In the broader literature, a case series study involving four cases, two of which involved military personnel, undertaken by Jackson et al. [23] found that deep or loaded squats performed while undertaking CrossFit activities were reported as the activity that gave rise to an acute injury of the articular cartilage of the patellofemoral joint. However, no other studies could be found that directly indicated that occupational activities involving loaded squatting were associated with an increased risk of acute articular cartilage tears. Another study, undertaken by Johnson-Nurse et al. [18], found that 65% of patients presenting for arthroscopy for investigation of mechanical symptoms of the knee had a history of acute trauma—typically involving twisting on an extended knee—and this may have implications for occupational knee injuries. Other studies by Loes et al. [20] and Arendt et al. [21] have found that certain sports have a higher prevalence of acute chondral lesions. Loes et al. [20] found that soccer had the highest prevalence of cartilage injuries in Swiss youth sports while Arendt et al. [21] found American intercollegiate soccer and basketball participation to be associated with cartilage injury. However, this latter study combined both meniscal and articular cartilage tears in their reporting.

Within the broader literature, there is strong evidence of acute articular cartilage tears occurring alongside concomitant injuries such as ACL ruptures [13,33,34,56,57], knee dislocations [58], or knee haemathrosis [11]. A retrospective population study undertaken in the United States by Wyatt et al. [59] found, in 261 patients presenting for primary ACL repairs, that 14.9% had accompanying articular cartilage injuries. Similarly, Burnett et al. [13] found that 7.8% of patients presenting for an ACL repair had an acute articular cartilage injury as well. Yuksel et al. [28], in their study of a Turkish military setting, found that initial knee trauma causing ACL tears was also associated with evidence of concomitant chondral injury. This study also found that there was an increase in the prevalence of chondral injury as time passed after the initial ACL tear, suggesting that joint instability leading to increased anterior translational shear may contribute to additional acute articular cartilage tears as well as increased risk of degenerative lesions occurring [28].

Other studies in non-human subjects have also identified potential risk factors related to acute articular cartilage tears. Castoldi et al. [45] undertook a cadaver study to evaluate whether ACL surgical repair techniques using cross-pins caused iatrogenic damage to the articular cartilage. When using a Rigidfix Cross Pin device, Castoldi et al. [45] found that the risk of chondral injury was high, with 80–100% of attempts having at least one pin penetrating the articular cartilage, resulting in acute damage. McCulloch et al. [47], in a study evaluating 48 intact porcine knee joints, found that both normal and elevated shear loads may increase the risk of chondrocyte damage in articular cartilage and reduce cartilage capacity to repair. In a study undertaken by Rudd et al. [46], the researchers performed dynamic dorsiflexion of the ankle in cadavers to determine whether impact loading, simulating a vehicle crash, resulted in specific injuries. Even though acute articular cartilage injuries were found in the majority of ankles, the authors indicated that further studies are required to determine the true mechanism and load tolerance of the cartilage for these types of injuries.

### Limitations

This review is not without its limitations. The eligibility criteria were designed to exclude all articles that were not undertaken on human participants, which may have meant that some studies were not included that might have provided contextual understanding regarding the mechanisms of injury associated with acute articular cartilage tears. The only eligible studies in this review were both pertaining to military settings and, therefore, the potential for extrapolation of findings to other physically demanding occupations is limited. Further studies are required to determine occupational risk factors associated with acute articular cartilage tears in other industries. Where univariate analyses were employed and adjustments for known confounders were not undertaken, the apparent relationships between risk factors and the development of the condition may be quite different from the actual relationships that would be evidenced if potential confounders were more comprehensively considered. Nevertheless, we have indicated where adjusted risk levels were provided in the included studies, and where these were not available, we have provided the best available evidence. Further research to develop adjusted estimates of risk is needed.

## 5. Conclusions

The findings from this review suggest that more physically active military populations may be at a higher risk of sustaining an acute articular cartilage tear than military populations that are less physically active. Specifically, the risk of an acute articular cartilage tear occurring has been observed to be higher in occupations that undertake more strenuous physical activity, such as Army or Marine occupations (and, particularly, enlisted ranks), than in individuals whose employment involves less strenuous physical activity. It is much less clear whether age, gender, or specific military training periods are associated with an increased risk of acute articular cartilage tears, although it appears that older military personnel, military women, and military trainees in their first six months of initial training may be at higher risk of experiencing this type of injury than their military counterparts. This review found a low level of evidence to suggest that exercise movements such as loaded squatting may also play a role in increasing the risk of acute articular cartilage tears. However, there is limited understanding of the specific mechanisms of acute articular cartilage injuries, with no identified studies being found that directly investigated the roles of specific occupational tasks or activities that may pose a risk for sustaining an acute articular cartilage tear. Future well-designed cohort studies that specifically evaluate the roles of occupational tasks and activities, as well as mechanisms of injury, in acute articular cartilage tears are needed to inform our understanding of how to mitigate the risks associated with acute articular cartilage tears.

## Figures and Tables

**Figure 1 healthcare-12-00595-f001:**
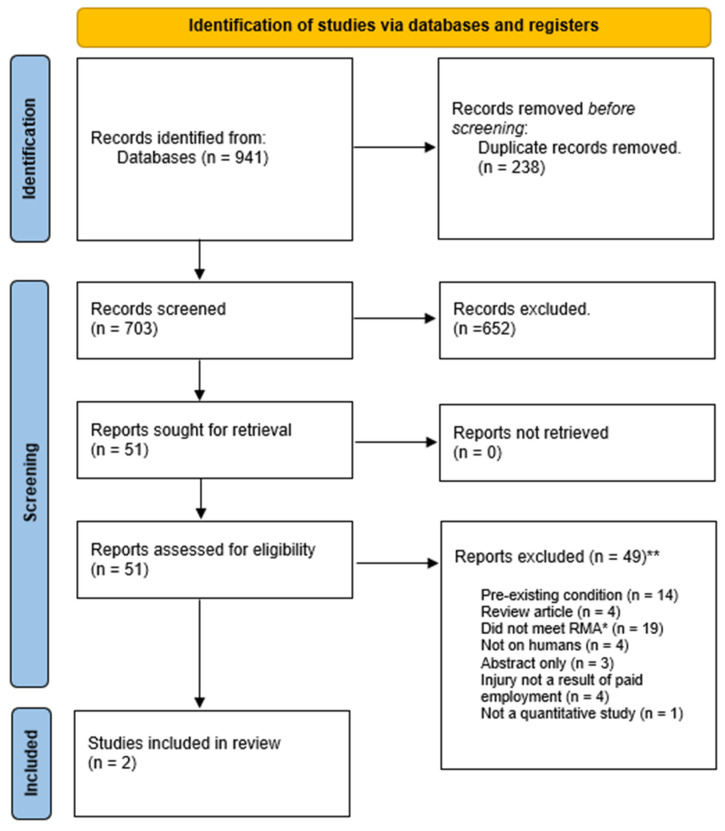
PRISMA diagram showing results of the search, screening, and selection processes [36]. * RMA: Australian Government Repatriation Medical Authority, which defined the diagnostic criteria for Acute Articular Cartilage Tear [38]. ** See Appendix A and Table A1 for excluded articles and references.

**Figure 2 healthcare-12-00595-f002:**
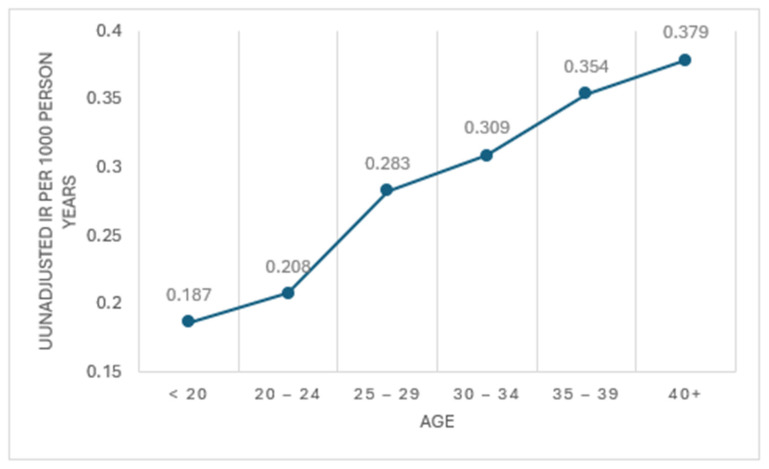
Unadjusted IR of Osteochondral Lesion of the Talus per 1000 person-years, by age, based on data extracted from Orr et al. [31].

**Table 1 healthcare-12-00595-t001:** Details of literature search including databases used, search terms, and filters.

Database	Search Terms
PubMed	(((“Acute articular cartilage tear” [Title/Abstract] OR “chondral tear” [Title/Abstract] OR “articular cartilage damage” [Title/Abstract]) OR “articular cartilage injury” [Title/Abstract] OR “chondral injury” [Title/Abstract] OR “hyaline injury” [Title/Abstract] OR “hyaline damage” [Title/Abstract])) AND (work*[Title/Abstract] OR occupation*[Title/Abstract] OR profession*[Title/Abstract] OR trade[Title/Abstract] OR employ*[Title/Abstract] OR military[Title/Abstract] OR defence[Title/Abstract] OR defense[Title/Abstract] OR airforce[Title/Abstract] OR “air force” [Title/Abstract] OR army[Title/Abstract] OR navy[Title/Abstract] OR recruit[Title/Abstract] OR soldier*[Title/Abstract] OR marines[Title/Abstract] OR “military personnel” [Title/Abstract])) AND (risk[Title/Abstract] OR predict*[Title/Abstract] OR prevalence[Title/Abstract] OR incidence[Title/Abstract] OR caus*[Title/Abstract] OR etiol*[Title/Abstract] OR frequenc*[Title/Abstract] OR rate*[Title/Abstract] OR mediat*[Title/Abstract] OR exposure*[Title/Abstract] OR likelihood[Title/Abstract] OR probability[Title/Abstract] OR “factor” [Title/Abstract] OR “factors” [Title/Abstract] OR “hazard” [Title/Abstract] OR “hazards” [Title/Abstract] OR predisposing[Title/Abstract])

**Table 2 healthcare-12-00595-t002:** Characteristics of included studies.

Authors, Year	Study Type	Setting	Participants	Sex Breakdown	Age (Years)	Study Aim	Outcome Measures of Interest for This Review	Study Quality Score
Orr et al. [31]	Cohort study	United States	Military personnel from across the armed forces	3207 Male 621 Female	<20 to over 40 years	To report the incidence rates and epidemiological variables related to osteochondral lesions of the talus (including—osteochondritis dissecans of the talus, transchondral talus fracture, and osteochondral talus fracture) in U.S. Armed Forces service members.	Incidence rates (and 95% CI) per 1000 person-years. Crude and adjusted incidence rates (and 95% CI) by strata for age, sex, race, rank, and service	89%
Kuikka et al. [32]	Cohort study	Finland	Military personnel undertaking basic training	Males only	18–30 years (Median 20 years)	To assess the incidence and risk factors for knee injuries leading to hospitalization.	Incidence rates (and 95% CI) per 1000 person-years. Odds ratios (and 95% CI) for specific risk factors	89%

CI—Confidence Intervals.

**Table 3 healthcare-12-00595-t003:** Key findings from included studies.

Authors, Year, Participants, Setting	Key Findings
Orr et al. [31] Military personnel from across the armed forces of the United States	*Incidence rates of first-time osteochondral lesions of the talus (including osteochondritis dissecans, transchondral talus fracture, and osteochondral talus)* Over the study period, there were 3828 first-time diagnoses of osteochondral lesion of the talus (OCLT) within a total population exposure period of 14,071,570 person-years, giving the following: Unadjusted overall incidence rate (IR) for OCLT: 0.27 first-time diagnoses per 1000 person-years of military service; Men: unadjusted IR 0.266 per 1000 person-years (reference group for adjusted incidence rate ratio [IRR]); Women: unadjusted IR 0.309 per 1000 person-years, adjusted IRR 1.34 (95% CI 1.23–1.47; adjusted for race, age, rank, and branch of military service); Black race: unadjusted IR 0.229 per 1000 person-years (reference group for adjusted incidence rate ratios [IRR]); White race: unadjusted IR 0.284 per 1000 person-years, adjusted IRR 1.53 (95% CI 1.40–1.68; adjusted for sex, age, rank, and service); Other race: unadjusted IR 0.270 per 1000 person-years, adjusted IRR 1.35 (95%CI 1.19–1.52; adjusted for sex, age, rank, and service); Junior Enlisted personnel: unadjusted IR 0.241 per 1000 person-years, adjusted IRR 1.92 (95%CI 1.66–2.21; adjusted for race, sex, age, and service); Junior Officer: unadjusted IR 0.192 per 1000 person-years (reference group for adjusted incidence rate ratios [IRR]); Senior Enlisted personnel: unadjusted IR 0.324 per 1000 person-years, adjusted IRR 1.61 (95%CI 1.42–1.84; adjusted for race, sex, age, and service); Senior Officer: unadjusted IR 0.286 per 1000 person-years, adjusted IRR 1.02 (95%CI 0.85–1.23; adjusted for race, sex, age, and service); Navy: unadjusted IR 0.182 per 1000 person-years (reference group for adjusted incidence rate ratios [IRR]); Air Force: unadjusted IR 0.196 per 1000 person-years, adjusted IRR 1.04 (95%CI 0.94–1.16; adjusted for race, sex, age, and rank); Marines: unadjusted IR 0.318 per 1000 person-years, adjusted IRR 2.03 (95%CI 1.81–2.28; adjusted for race, sex, age, and rank) Army: unadjusted IR 0.380 per 1000 person-years, adjusted IRR 2.16 (95%CI 1.97–2.36; adjusted for race, sex, age, and rank); Age < 20 years: unadjusted IR 0.187 per 1000 person-years (reference group for adjusted incidence rate ratios [IRR]); Age 20–24 years: unadjusted IR 0.208 per 1000 person-years, adjusted IRR 1.22 (95%CI 1.05–1.42; adjusted for race, sex, rank, and service); Age 25–29 years: unadjusted IR 0.283 per 1000 person-years, adjusted IRR 1.95 (95%CI 1.66–2.30; adjusted for race, sex, rank, and service); Age 30–34 years: unadjusted IR 0.309 per 1000 person-years, adjusted IRR 2.36 (95%CI 1.98–2.82; adjusted for race, sex, rank, and service); Age 35–39 years: unadjusted IR 0.354 per 1000 person-years, adjusted IRR 2.89 (95%CI 2.40–3.47; adjusted for race, sex, rank, and service); Age 40+ years: unadjusted IR 0.379 per 1000 person-years, adjusted IRR 3.34 (95%CI 2.76–4.05; adjusted for race, sex, rank, and service). (See Figure 2) *Engagement in active conflict and associated high-demand physical activities:* The authors noted that within the study period 1999–2008, a continuous significant increase in the OCLT incidence rates was observed from 2002 to 2008, with a 2008 IR of 0.556 per 1000 person-years and an unadjusted IRR of 3.41 (95% CI 3.39 to 3.43) when the IR for 2008 was compared to the IR of 0.163 per 1000 person-years for 2002. The authors further noted that this period of steady increase in OCLT IR corresponded directly to years in which active-duty military personnel were actively engaged in military conflict in Iraq and Afghanistan in the global war on terrorism, and hence in increasingly higher demand physical activities. The authors also acknowledged that more aggressive and early use of radiographic modalities such as magnetic resonance imaging as well as increased provider recognition of this diagnosis might have contributed to these findings of increasing IR from 2002 to 2009.
Kuikka et al. [32] Male military personnel undertaking basic training in Finland	Total number of knee injuries of all types leading to at least one hospitalisation: 1073 (0.8% of all male conscripts undertaking training) and acute articular cartilage lesions were diagnosed in 18 of the hospitalised knee injury cases (1.7% of all knee injuries). *Incidence rate of acute articular cartilage injuries of the knee resulting in hospitalisation* Male incidence rate: 0.2 acute articular cartilage lesions of the knee resulting in hospitalisation per 1000 person-years, 95% CI (0.1–0.3). *Timing of acute articular cartilage lesions of the knee * 10% of hospitalised acute articular cartilage tears occurred during the basic training period (first 8 weeks, days 1–56 of initial training); 60% of hospitalised acute articular cartilage tears occurred during the special and team training period (next 18 weeks, days 57 to 180 of initial training); 30% of hospitalised acute articular cartilage tears occurred during the leadership period (final 26 weeks, days 180 to 365 of initial training).

IR—Incidence rate; IRR—Incidence rate ratio; CI—Confidence interval.

## Data Availability

All data generated or analysed during this study are included in this published article.

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
