# Peer review of "Incidence and Risk Factors for Acute Articular Cartilage Tears in Military and Other Occupational Settings: A Systematic Review"

_healthcare, 2024, doi:10.3390/healthcare12050595_

Round 1

Reviewer 1 Report

Comments and Suggestions for Authors

The study hypothesis should be clearly defined in the Introduction.

Author Response

Reviewer

Feedback

Response

Reviewer 1

The study hypothesis should be clearly defined in the introduction

Thank you for this suggestion. As the review is exploratory, it is not appropriate to state a hypothesis in the introduction, as the findings should reflect the literature considered in the review. We did not set out to test a specific hypothesis but rather to identify and report all evidence relevant to the stated aim of the review.

Reviewer 2 Report

Comments and Suggestions for Authors

 - I suggest the authors replace the first and second paragraph of the introduction with a more viable introductory phrase that target and refers to military forces/ occupational settings

- detail on the the discussion on how specific movements or occupational activities contribute to the risk of injury.

- Broaden the introduction to include other physically demanding occupations, such as construction workers or athletes in non-contact sports

- underline the lack of studies on non-sporting and non-military occupational risks for acute articular cartilage injuries

- add time frame restrictions to your eligibility criteria to focus on the most relevant and contemporary evidence.

- mention why NHMRC was considered more suitable than GRADE for your review

- Clearly delineate between what is known from military studies and what can be inferred or hypothesized about other occupational settings

Author Response

Reviewer

Feedback

Response

Reviewer 2

Suggest replace the first and second paragraph of the introduction with a more viable introductory phrase that target and refers to military forces/occupational settings.

We have added an additional sentence to the end of the first paragraph to highlight this point. See lines 40-42

Detail on the discussion on how specific movements or occupational activities contribute to the risk of injury

We have added additional information to the third paragraph to highlight this point. See lines 59-60

Broaden the introduction to include other physically demanding occupations, such as construction workers or athletes in non-contact sports

We have added additional text and references to include other physical demanding occupations such as construction and mining and have added additional references to support this. See lines 40-41.

Underline the lack of studies on non-sporting and non-military occupational risks for acute articular cartilage injuries

We have added this text in the background section. See lines 79-80

Add time frame restrictions to your eligibility criteria to focus on the most relevant and contemporary evidence

We chose not to restrict the date range for this review, given the limited literature that has focused on this topic, as we wanted to ensure that all relevant articles were included. However, it should be noted that the only eligible articles considered in the review were published in 2011 and 2013 which are relatively recent years. As highlighted in the review, the lack of more contemporary studies in this area warrants further research.  

Mention why NHMRC was considered more suitable than GRADE for your review

The paper outlines a justification as to why NHMRC was considered more suitable, in the paragraph under section 2.5. Level of evidence and methodological quality assessment.

As highlighted in the text the specific reason is listed on lines 139-144. “For this systematic review, the NHMRC scale was selected rather than the Grading of Recommendations Assessment, Development, and Evaluation (GRADE) system of rating quality of evidence and grading strength of recommendations in systematic reviews. This was because the GRADE is not designed for assessing evidence associated with studies that focus on answering research questions about risk or prognosis.”

Clearly delineate between what is known from military studies and what can be inferred or hypothesized about other occupational settings.

Only two eligible studies, both pertaining to military occupations were identified for inclusion in our review and no studies were identified which focused on other occupations. On this basis, all current knowledge of these injuries in occupational settings is limited to military occupational settings and any extrapolation of these findings to other occupational settings would need to be validated with further research. We have, however, added an additional sentence in the limitations section to highlight this. See lines 400-403.

Reviewer 3 Report

Comments and Suggestions for Authors

The manuscript "Incidence and risk factors for acute articular cartilage tears in military and other occupational settings: A systematic review" is intended to identify the aetiology, incidence, and known risk factors for workers sustaining acute articular cartilage tears in paid occupational settings. While the topic is interesting, the approach lacks the technical soundness. The following comments needs to be addressed.

1. The introduction is poorly written. All the references in the introduction are very old. the latest article referred is from 2019.Authors need to do through review of the recent developments. 

2. The representation of the results section is very poor. Interestingly, there is not a single image in the entire section. Graphical representations of the statistics and facts will be very effective. This has to be included.

3. The information should also be presented in tabulated formats for better clarity.

4. The general standard of English displayed is low. Suggest to recheck the entire manuscript for grammatical an sentence construct errors.

5.Rewrite the conclusions in point-wise.

Comments on the Quality of English Language

The general standard of English displayed is low. Suggest to recheck the entire manuscript for grammatical an sentence construct errors.

Author Response

Reviewer

Feedback

Response

Reviewer 3

The manuscript "Incidence and risk factors for acute articular cartilage tears in military and other occupational settings: A systematic review" is intended to identify the aetiology, incidence, and known risk factors for workers sustaining acute articular cartilage tears in paid occupational settings. While the topic is interesting, the approach lacks the technical soundness. The following comments needs to be addressed.

Thank you for this comment. The review was reported in accordance with the PRISMA guidelines and was conducted in a systematic fashion, consistent with technical advice for contemporary systematic reviews. We have addressed specific additional comments from the reviewer below.

The introduction is poorly written. All the references in the introduction are very old. the latest article referred is from 2019.Authors need to do through review of the recent developments. 

We have made some amendments to the introduction in response to specific feedback from reviewer 2 and we believe the introduction now meets expectations for a sound background to the study.

We have undertaken another review of the literature and there have been no further recent references to cite that specifically focus on acute articular cartilage tears.

The representation of the results section is very poor. Interestingly, there is not a single image in the entire section. Graphical representations of the statistics and facts will be very effective. This has to be included.

We have reviewed the results section and the only data that would be appropriate to represent as a figure, based on it indicating a trend, is the data related to incidence rates of OCLT by age category. We have updated the results section to include this figure (See Figure 2 line 241). All other data were categorial, without evidence of trends across multiple categories, and so these data are best represented in tabular form.

The information should also be presented in tabulated formats for better clarity.

Key information is presented in Tables 2 and 3. The tables presented in the draft reviewed by the reviewers is formatted slightly differently to the format we submitted and we will work with the editors through the proofing phase to ensure the published tables are appropriate and clear.

The general standard of English displayed is low. Suggest to recheck the entire manuscript for grammatical an sentence construct errors.

The first language of all authors is English. We have rechecked the manuscript and we have not identified any remaining issues following amendments as indicated above. We also note that the three other reviewers had no concerns with the English grammar and expression in the paper.

Rewrite the conclusions in point-wise.

We have presented the conclusion in the format required by the journal and the paragraph structure involves a series of points made in sentence format. We note that the other three reviewers had no concerns with the format for presentation of the conclusion but we are happy to discuss this further with the editor if required.

Reviewer 4 Report

Comments and Suggestions for Authors

The exciting part of this review is the investigations of military and other sports-related occupations with acute articular cartilage tears, as the title suggests. These include explanations of which military branches are more prone to injuries and which sports activities are more prone to injuries, and also provide explanations based on age, gender, and ranks. However, instead of simply stating a retrospective articles, the author should have included more of your own conclusions. This gave me the impression that there were many limitations in these areas of research due to the lack of previous research and the ambiguity of much of it. The final conclusion is too predictable. To have the article published, the authors should address the following concerns.

This review focuses on acute articular cartilage tears and excludes chronic or degenerative articular cartilage tears. However, I feel it is necessary to provide the reader with more information about the previous connections and differences between the two, rather than simply saying that because you are aiming at the former, the latter is all but ruled out. The same problem is as written in the limitation that only human studies were included all non-human studies were excluded, which the authors themselves emphasized as a deficiency of this systematic review.

I didn't feel like I gained much from reading this review. It appears that the author followed a systematic approach with strict screening conditions, resulting in the exclusion of numerous articles. The focus of the review was primarily on explaining the screening process in the results section. To maintain a better balance, the discussion session could be condensed. Alternatively, the authors can incorporate the key content into the results section.

In Figure 1, what does the blue double underline underneath "removed" mean? If it has a special meaning, please explain it in the figure legend.

You mentioned acute joint cartilage tears associated with concurrent injuries, however, only ACL injuries were discussed in detail. Please provide more information about other injuries. I suggest including meniscal tears as well.

Author Response

(The authors gave the same response as above.)

Round 2

Reviewer 4 Report

Comments and Suggestions for Authors

NA